# Is the burden of anaemia among Indian adolescent women increasing? Evidence from Indian Demographic and Health Surveys (2015–21)

**Mahashweta Chakrabarty**[1], **Aditya Singh**[1,2]*, **Shivani Singh**[3], **Sourav Chowdhury**[4]

**1** Banaras Hindu University, Varanasi, Uttar Pradesh, India, **2** Girl Innovation, Research, and Learning (GIRL) Center, Population Council, New York, NY, United States of America, **3** Indian Health Action Trust, Lucknow, Uttar Pradesh, India, **4** Raiganj University, Raiganj, West Bengal, India

* adityasingh@bhu.ac.in

## Abstract

Anaemia is a significant public health issue, particularly affecting women in India. However, little is known about the burden of anaemia among adolescent women in India over time. This study aimed to analyse the change in the prevalence of anaemia among adolescent women in India from 2015 to 2021 and identify the factors associated with anaemia in this population. This study used information on 116,117 and 109,400 adolescent women (aged 15–19) from the fourth and fifth round of National Family Health Survey, respectively. Bivariate statistics and multivariable logistic regression were employed to identify the statistically significant predictors of anaemia. The prevalence of anaemia among adolescent women in India increased from 54.2% (99% CI: 53.6–54.8) to 58.9% (99% CI: 58.3–59.5) over the study period (2015–16 to 2019–21). Among the 28 Indian states, 21 reported an increase in the prevalence of anaemia. However, the levels of increase varied across the states. While Assam, Chhattisgarh, and Tripura showed a substantial rise of 15 percentage points, the states of Punjab, Karnataka, Telangana, Bihar, and Madhya Pradesh recorded a marginal increase of less than 5 percentage points. Notably, Uttarakhand and Kerala exhibited a decline in anaemia prevalence during the study period. Additionally, the number of states with anaemia prevalence exceeding 60%, doubled from 5 in 2015–16 to 11 in 2019–21. Several factors were found associated with anaemia, including having more than one child (AOR: 1.33, 99% CI: 1.16–1.51), having no education (AOR: 1.25, 99% CI: 1.16–1.34), belonging to Scheduled Tribes (AOR: 1.47, 99% CI: 1.40–1.53), being in the lowest wealth quintile (AOR: 1.17, 99% CI: 1.12–1.23), year of survey (AOR: 1.26, 99% CI: 1.23–1.29), and being underweight (AOR: 1.10, 99% CI: 1.07–1.12). In conclusion, the rise in anaemia prevalence among adolescent women in India suggests the need for targeted interventions to mitigate the burden of anaemia and enhance the overall health of this population.

**Data Availability Statement:** The National Family Health Survey-5 dataset used in the study is publicly available at the official website of

Demographic and Health Surveys (DHS) https://dhsprogram.com/data/available-datasets.cfm. Anyone can obtain this data by making a formal request to the DHS.

**Funding:** The authors received no specific funding for this work.

**Competing interests:** The authors have read the journal's policy and have the following competing interest: AS is a member of the PLOS Editorial Board outside of the submitted work. This does not alter our adherence to PLOS policies on sharing data and materials.

## Introduction

According to the World Health Organization (WHO), anaemia is a disorder in which the number of red blood cells or haemoglobin concentration within the red blood cells is below normal which subsequently results in the decreased oxygen-carrying capacity of blood [1]. Haemoglobin is an iron-containing protein in the red blood cells (RBC) that transports oxygen from the lungs to the tissues and carries carbon dioxide from tissues back to lungs [2]. The amount of haemoglobin in whole blood is expressed in grams per decilitre (g/dl). Anaemia is defined as a haemoglobin concentration below a specified cut-off point which depends on individual's age, gender, physiological status, smoking habits, etc. The WHO defines anaemia in pregnant women as a haemoglobin concentration <11.0 g/dl, and anaemia in non-pregnant women as a haemoglobin concentration <12.0 g/dl [2].

Globally anaemia is responsible for approximately 50.3 million years of healthy life lost due to disability (YLDs) in 2019, with highest YLDs attributable to anaemia coming from South Asia, Western Sub-Saharan Africa, and Eastern Sub-Saharan Africa [3]. Women are susceptible to anaemia of low haemoglobin concentration due to their unique physiological needs, including menstrual blood loss, and pregnancy [4–6]. Therefore, the burden of anaemia is disproportionately higher in women as compared to men. This is particularly true for low- and middle-income countries [2]. In these countries, anaemia is a major cause of morbidity among women of reproductive age group, particularly adolescents [7, 8].

Anaemia can have significant consequences on physical and mental health, and economic and social well-being of any woman [7]. The most common physical symptoms of anaemia include fatigue, weakness, and shortness of breath, which can lead to depression, anxiety, and decreased quality of life, making it difficult for women to perform daily activities and work [9–11]. Anaemia can also cause headaches, pale skin, irritability, and a rapid heartbeat [12]. Furthermore, anaemia in pregnant women is associated with an increased risk of maternal mortality, preterm birth, low birth weight, and developmental delays in the baby [13]. Studies have also shown that anaemia is associated with poor cognitive function and memory problems [14, 15]. These physical and cognitive deficits resulting from anaemia have a significant impact on economic development of a country [16, 17]. Previous studies suggest that the physical and cognitive deficits resulting from iron deficiency anaemia cost developing countries up to a 4.05% loss in gross domestic product (GDP) per year [18]. In absolute dollar terms, the losses in South Asia are significant, with close to $4.2 billion lost annually in Bangladesh, India, and Pakistan. The losses associated with iron deficiency anaemia amount to about 1.2% of GDP in India [19].

Adolescent women, who constitute about 17% of the total female population in India, are particularly vulnerable to anaemia due to their unique physical and physiological changes that occur during this stage of life [20, 21]. Adolescence is termed as "second sensitive developmental period" after early childhood, as it is a time of rapid growth and development, which can lead to increased nutritional demands and a higher risk of nutritional deficiencies [22–24]. Adolescent girls are at a higher risk of iron deficiency anaemia due to factors like iron loss during menstruation and inadequate iron intake [25, 26]. They may also face risks of other nutrient deficiencies like vitamin B12 and folate, which are important for red blood cell production [26, 27]. Additionally, health conditions like chronic diseases and infections can contribute to anaemia in adolescent women [21, 28]. Factors such as limited decision-making power, financial autonomy, and restricted healthcare access may exacerbate these risks [24, 29]. Addressing anaemia among adolescent women is a key step to ensure that they have adequate opportunities to reach their full potential and enjoy their human rights [31]. Treating anaemia in adolescent girls is crucial for achieving United Nation's Sustainable Development Goal 3, which aims

to promote healthy lives and well-being for all, and the Global Nutrition Targets 2025 set by the World Health Organization [32, 33].

Several studies have investigated the factors that contribute to the high prevalence of anaemia among women and adolescent girls in developing countries [29–32]. Evidence suggests that poverty, which limits individuals access to nutritious food and healthcare, is a major determinant of anaemia these countries [33]. It has been noted that adolescent females in poorer nations often have inadequate nutritional consumption, and deficiencies in key micronutrients such as iron, folic acid, and vitamin B12 contribute to the significant risk of anaemia [8, 25, 28, 31, 33–35]. This is worsened by societal views that emphasize the dietary demands of men and boys above those of women and girls [36, 37]. Women with lower levels of education may have less knowledge about nutrition and may be more likely to engage in behaviours that increase their risk of anaemia, such as consuming a diet that is low in iron [20, 38]. Pregnancy and delivery can contribute to women's elevated anaemia rates due to blood loss after birth as well as the increased nutritional needs of pregnancy [5, 39]. Aside from these physiological elements, several socioeconomic and cultural factors also have a role in the development of anaemia in women [33].

In India, several previous studies have specifically addressed the issue of anaemia among adolescent women. However, these studies have been limited in their scope, focusing only on small geographical areas [7, 29]. Consequently, the findings of these studies cannot be generalized to a larger population, impeding a comprehensive understanding of anaemia prevalence in the country. Also, none of the previous studies has examined how the burden of anaemia among adolescent women (aged 15–19) has decreased or increased at the national and state level over the years. This gap in knowledge highlights the need for a comprehensive analysis of anaemia trends within this specific demographic group.

Therefore, the specific aim of this study is to examine the change in the prevalence of anaemia among adolescent women in India and its states between 2015–16 and 2019–21. In addition, this study also aims to examine various biodemographic, socioeconomic, behavioural, geographic, and health-related factors associated with anaemia among adolescent women in India. To accomplish these aims, this study uses cross-sectional data from the two latest rounds of the National Family Health Survey (NFHS). The findings from this study are expected to provide valuable insights for policymakers and program implementers, enabling the formulation of targeted strategies to address anaemia and mitigate its impact on adolescent women in India.

## Data and methods

### Data source

The study utilizes the most recent rounds of NFHS, namely NFHS-4 and NFHS-5, conducted in 2015–16 and 2019–21, respectively. These are large-scale nationally representative surveys offering comprehensive data on a wide range of topics including demography, family planning, reproductive health, nutrition, and various other health-related issues [40, 41]. The NFHS-4 (2015–16) was designed to provide the estimates of all the key indicators at the national, state (29 states and 6 union territories), and district levels (for 640 districts), and NFHS-5 was designed to provide information for 707 districts, 28 states, and 8 union territories. The survey adopted a uniform sample design for all the states of the country. For both the rounds of surveys, in each state, a two-stage stratified sampling procedure was employed to select samples from rural and urban areas, respectively. The two-stage sample selection for the rural areas encompassed selecting villages, which served as Primary Sampling Units (PSUs), with Probability Proportional to Size (PPS) at the first stage, followed by the random selection

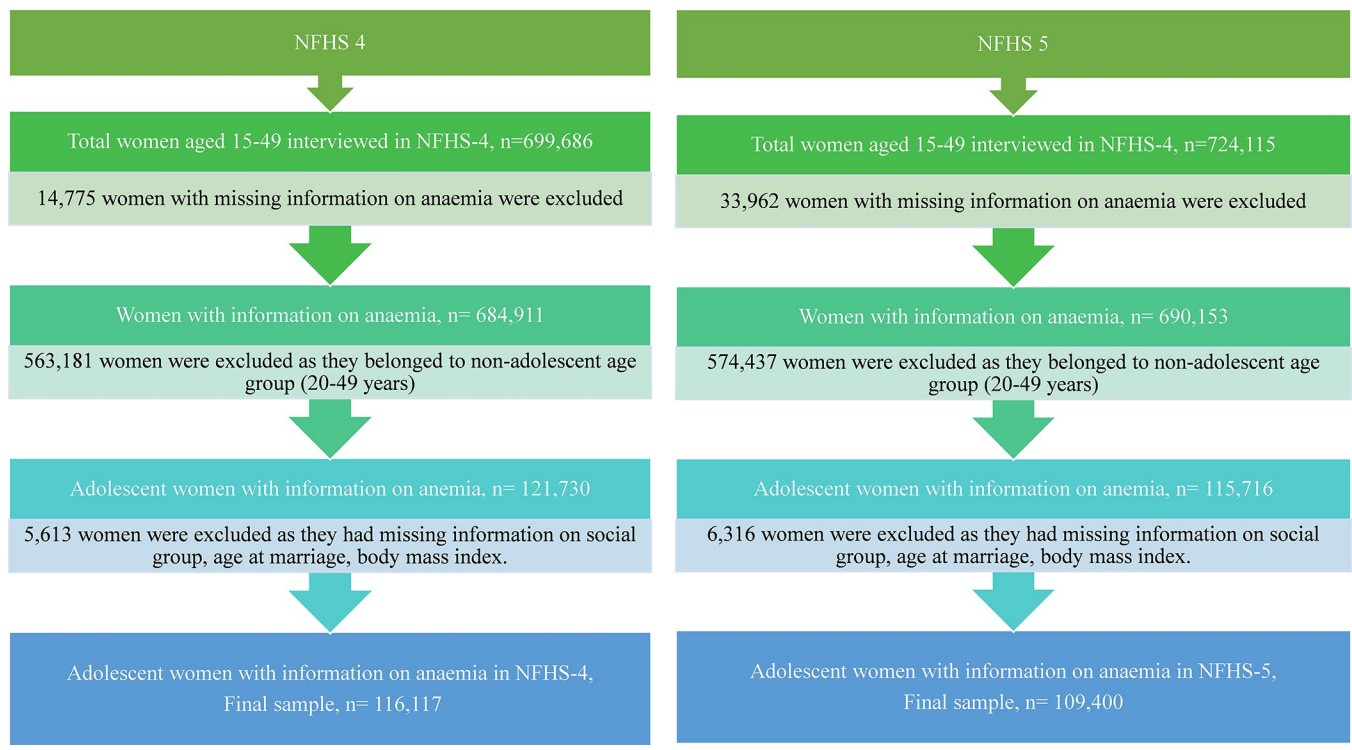

**Fig 1. Selection of sample size for the current study using NFHS-4 and NFHS-5.** Source: Authors' own creation.

of households within each PSU in the second stage. However, wards were selected with PPS sampling in the first stage for the urban areas, followed by randomly selecting one Census Enumeration Block (CEB) from each sampled ward. Finally, households were randomly selected within each selected CEB. The details of sampling for both surveys are available in their respective national reports which are available online at https://dhsprogram.com/data/available-datasets.cfm. NFHS-4 interviewed 699,686 women of reproductive age (15–49 years) from 601,509 households and NFHS-5 interviewed 724,115 women from 636,699 households, with a response rate of 97%. To obtain our study sample, at first, we extracted data only for those adolescent women for whom information regarding anaemia was available, and then excluded observations with missing information on some variables, from both the datasets. Our final sample comprised 116,117 and 109,400 adolescent women aged 15–19 years from NFHS-4 and NFHS-5, respectively. The process of obtaining the analytical sample is detailed in Fig 1.

## Ethics statement

Ethical permission was not required as the data is taken from an open domain. The survey data used in this study can be obtained by making a formal request on the official website https://dhsprogram.com/data/new-user-registration.cfm.

## Testing of anaemia

In both NFHS-4 and NFHS-5, finger stick blood samples were collected from all women aged 15–49 who voluntarily consented to the testing [42]. A drop of blood from a finger prick was collected in a microcuvette to obtain blood samples followed by an on-site haemoglobin test

which was carried out using a battery-powered portable HemoCue Hb 201+ analyzer [40, 41]. The results of the anaemia test were then recorded in the NFHS-4 Biomarker Questionnaire and provided immediately to each individual tested both verbally and in writing.

## Variables

**Dependent variable.** The variable representing the level of anaemia in both the NFHS-4 and NFHS-5 datasets is an ordinal variable having four categories: no anaemia, mild anaemia, moderate anaemia, and severe anaemia. For nonpregnant women, mild anaemia is defined as hemoglobin levels between 11.0 to 11.9 g/dl, while for pregnant women, it is defined as hemoglobin levels between 10.0 to 10.9 g/dl. Moderate anaemia is defined as hemoglobin levels between 8.0 to 10.9 g/dl for nonpregnant women, and 7.0 to 9.9 g/dl for pregnant women. Severe anaemia is defined as hemoglobin levels of 8.0 g/dl or less for nonpregnant women, and 7.0 g/dl or less for pregnant women [41].

To analyze the association between anaemia and various factors among reproductive women, we created a binary outcome variable from the four categories of anaemia in the NFHS-5 dataset. Women with either mild, or moderate, or severe anaemia were considered "to have anaemia" and were coded as "1". Women with no anaemia were coded as "0" or "women do not have anaemia". Our main aim was to identify whether women were anaemic or not anaemic, and this binary variable helped us to categorize women into these two groups.

**Independent variables.** This study has used a set of independent variables to examine the factor associated with anaemia prevalence among adolescent women in India. The choice of variables included in the analysis as independent variables was informed by previous literature available on anaemia [7, 21, 43, 44]. We thoroughly search various databases including PubMed, Web of Science, and Google Scholar for relevant literature and reviewed them to extract potential predictors of anaemia for our analysis. We divided these variables into the following major groups: biodemographic, socioeconomic, geographic, behavioural, and health-related determinants [44, 45]. The following variables were included in the analysis: marital status (not married, married before 18 years, married 18 years or above), parity (no child, single child, two and more), pregnancy and breastfeeding status (pregnant, breastfeeding not breastfeeding/not pregnant), education (no education, primary, secondary, higher), social groups (Scheduled Caste or SC, Scheduled Tribe or ST, Other Backward Classes or OBC, Others), religion (Hindu, Muslim, Christian, others), household wealth index (poorest, poorer, middle, richer, richest), place of residence (rural, urban), region of residence (northern, central, western, eastern, southern, north-eastern) mass media exposure (no exposure, low, medium, high), dietary habit (vegetarian and non-vegetarian), contraceptive use (not using, traditional method, modern method), alcohol consumption (no, yes), consumption of tobacco (no tobacco, uses tobacco: smoke or smokeless), body mass index (underweight, normal weight, overweight, obesity), diabetes (no, yes), amenorrhea (no, yes). A detailed description of each independent variable including its categorization is given in Table 1.

A conceptual framework showing the potential independent variables associated with anaemia is provided in Fig 2.

## Statistical analysis

The statistical analysis involved several steps to assess changes in anaemia prevalence over time in India and its states, and to identify factors associated with anaemia using multivariable binary logistic regression.

In the initial stage, we presented the distribution of the sample of adolescent women across various background characteristics. Following that, we calculated the prevalence of anaemia

**Table 1. Description of independent variables.**

| Independent variables | Description |
|---|---|
| *Biodemographic and socioeconomic factors* | |
| Marital status | Marital status variable was created by dividing the age at marriage into three categories. The first category was 'not married' (coded as 0). The second category comprises the women who were married off before the legal age at marriage i.e., 18 years (coded as 1), and third category comprised women who were married on or after 18 years of age (coded as 2). |
| Parity | Parity refers to the number of live births a woman has had up to a certain point in time. This variable had three categories: 'no children' (coded as 0), 'single child' (coded as 1), and '2 or more children' (coded as 2). |
| Pregnancy and breastfeeding status | Pregnancy and breastfeeding status had three categories. Women who were pregnant at the time of interview were coded as 1. Women who were breastfeeding at the time of interview were coded as 2. Those women who were neither pregnant nor breastfeeding at the time of interview were coded as 3. |
| Level of education | Education level of women indicates the highest level of education of the respondent. This variable was classified into four categories: 'no education' (coded as 0), 'primary' (coded as 1), 'secondary' (coded as 2), 'higher' (coded as 3). |
| Social group | The "social group" variable in our study refers to the categorization of individuals into different social groups based on the official categorization adopted by the Government of India. The four categories included in this variable are Scheduled Castes (SC) (coded as 1), Scheduled Tribes (ST) (coded as 2), Other Backward Classes (OBC), and Others (coded as 4). This categorisation of Indian population into four broad social groups has evolved over time and has its roots into India's caste system, a rigid and discriminatory social stratification system, primarily governs the majority of the population. The communities at the bottom of the caste-based social hierarchy have been subjected to centuries of discrimination, oppression, and abuse from those higher up in the hierarchy. These communities are known as Scheduled Castes. They are officially recognised in the Constitution of India and listed in various government orders. Other Backward Classes (OBCs) are also socioeconomically and educationally disadvantaged communities, but their position in the social hierarchy is slightly higher than SCs who are placed at the bottom of the caste hierarchy. The term 'backward' refers to historically disadvantaged groups in India, that are considered socially and educationally disadvantaged. The term has been officially recognized by the Indian government as part of its affirmative action policies to address historical social and economic inequalities in the country. The term 'backward' in this context does not imply a value judgment, but rather refers to the historical social and economic disadvantages that these groups have faced. The remaining caste groups, which form the hierarchy's top order, are grouped into a residual category called 'Others'. Scheduled Tribes (STs), on the other hand, include tribal communities which are characterised by their unique cultural practices and geographical isolation rather than their placement in the caste hierarchy. STs continue to be one of India's most marginalised and disadvantaged communities, having faced several social, economic, and political challenges throughout history. SC, ST, and OBC communities are recognized and protected under the Indian Constitution, and the Constitution of India provides for their protection and advancement. The National Commission for Scheduled Castes, National Commission for Scheduled Tribes, and National Commission for Backward Classes maintain the list of these communities [46, 47]. |
| Religion | In our study, religion was divided into four major categories: 'Hindu' (coded as 1), 'Muslim' (coded as 2), 'Christian' (coded as 3), and 'Others' (coded as 4). Others category comprised the Sikh, Buddhist/neo-Buddhist, Jain, Jewish, Parsi/Zoroastrian religions. |

(*Continued*)

**Table 1.** (Continued)

| Independent variables | Description |
|---|---|
| Household wealth | Household wealth was determined based on household scores assigned using principal component analysis. These scores were calculated based on various indicators, such as consumer goods owned and housing characteristics like toilet facilities, flooring materials, and source of drinking water. To determine the national wealth quintiles, the scores assigned to each household member were ranked and then divided into five categories: 'poorest' (coded as 1), 'poorer' (coded as 2), 'middle' (coded as 3), 'richer' (coded as 4), and 'richest' (coded as 5). Each category contained 20 percent of the population, and these categories were used as cut-offs to classify households into different wealth quintiles. It is important to note that the wealth index used in our study was derived from the NFHS-5 dataset, and was not created by us (authors). |
| *Geographical factors* | |
| Place of residence | This variable had two categories: 'urban' (coded as 1), and 'rural' (coded as 2). |
| Region of residence | To construct this variable, Indian states and UTs are grouped into six categories. 'Northern' (coded as 1) includes Jammu & Kashmir, Ladakh, Himachal Pradesh, Punjab, Rajasthan, Haryana, Uttarakhand, Chandigarh (Union Territory—UT) and Delhi; 'central' (coded as 2) includes the states of Uttar Pradesh, Madhya Pradesh and Chhattisgarh; 'eastern' (coded as 3) includes the states of Bihar, Jharkhand, West Bengal and Odisha; 'western' (coded as 4) includes the states of Gujarat, Maharashtra, Goa and UTs of Dadra & Nagar Haveli and Daman & Diu; 'southern' (coded as 5) includes the states of Kerala, Karnataka, Andhra Pradesh, Tamil Nadu and the UTs of Andaman & Nicobar Islands, Pondicherry and Lakshadweep); 'northeastern' (coded as 6) includes the states of Sikkim, Assam, Meghalaya, Manipur, Mizoram, Nagaland, Tripura, and Arunachal Pradesh. This classification is used in NFHS-5 India report. |
| *Behavioral factors* | |
| Mass media exposure | In NFHS-5, women were asked three questions related to their media exposure: 1. How often do they read newspapers/magazines? 2. How often do they watch television? 3. How often do they listen to the radio? The response options provided were: 'at least once a week', 'less than once a week', and 'not at all'. These options were dichotomised into 'yes' or 'no' to determine whether the respondent read newspapers/magazines, watched television, and listened to the radio. Then combining the responses of these three dichotomous variables, we created a new variable to determine the level of mass-media exposure for each respondent. It had four categories as follows:<br>'No exposure' (coded as 0): This category included respondents who reported not being exposed to any form of mass media.<br>'Low exposure' (coded as 1): This category included respondents who reported being exposed to only one type of mass media (e.g., either newspapers/magazines, television, or radio).<br>'Medium exposure' (coded as 2): This category includes respondents who reported being exposed to any two types of mass media.<br>'High exposure' (coded as 3): This category includes respondents who reported being exposed to all three types of mass media. |
| Vegetarianism | Vegetarianism had two categories. If a woman never ate fish, egg, or chicken/meat they were termed as 'vegetarian' (coded as 1), and if a woman reported eating fish, egg, or chicken daily, weekly, or occasionally, then they were termed as 'non-vegetarian' (coded as 2). |
| Alcohol consumption | Alcohol consumption had two categories: 'no' (coded as 0) and 'yes' (coded as 1). |
| Consumption of tobacco in any form | This variable had two categories, i.e., 'uses tobacco: smoke or smokeless form' (coded as 1), and 'no' use of tobacco (coded as 0). |
| Current contraceptive use | The question on current contraceptive use in both NFHS-4 and NFHS-5 included a comprehensive range of 20 contraceptive methods in the response options. Women who reported no use of contraception were coded as 0. Use of periodic abstinence, prolonged abstinence, withdrawal, or other traditional methods were considered as 'traditional methods' of contraception (coded as 1). Male and female sterilization, injectables, intrauterine devices (IUDs/PPIUDs), contraceptive pills, implants, female and male condoms, diaphragm, foam/jelly, the standard days method, the lactational amenorrhea method, and emergency contraception were categorized as 'modern methods' of contraception (coded as 2). |

*(Continued)*

**Table 1.** (Continued)

| Independent variables | Description |
|---|---|
| *Health related factors* | |
| Body mass index | The classification of body mass index (BMI) was based on the guidelines set by the World Health Organization (WHO). It is defined as a person's weight in kilograms divided by the square of the person's height in meters ($kg/m^2$). Body mass index is divided into four categories: 'underweight' ($<18.5$ $kg/m^2$) (coded as 1), 'normal' (18.5–24.99 $kg/m^2$) (coded as 1), 'overweight' (25–32 $kg/m^2$) (coded as 2), and obese ($>32$ $kg/m^2$) (coded as 3). |
| Currently has diabetes | Self-reported diabetes had two categories: 'no' (coded as 0) 'yes' (coded as 1). |
| Currently amenorrheic | Amenorrhea is the absence of menstruation, typically defined by the absence of one or more menstrual cycles. Self-reported amenorrhea had two categories: 'no' (coded as 0), 'yes' (coded as 1). |
| Year | Year had two categories: 'NFHS 4' conducted in 2015–16 (coded as 0), 'NFHS 5' conducted in 2019–21 (coded as 1). |

among these adolescent women and examined how it varied based on their respective background characteristics. This analysis was performed separately for both the survey periods of 2015–16 and 2019–21, enabling us to observe any changes or trends in anaemia prevalence over time within specific subgroups defined by their background characteristics.

To assess the temporal change in anaemia prevalence during the study period, we initially calculated the prevalence of anaemia at the national level. However, as national-level analysis may obscure regional variations, we proceeded to calculate the prevalence of anaemia for each individual state during both time periods. This allowed us to analyse the state-level changes in anaemia prevalence between 2015–16 and 2019–21. By examining the state-level data, we

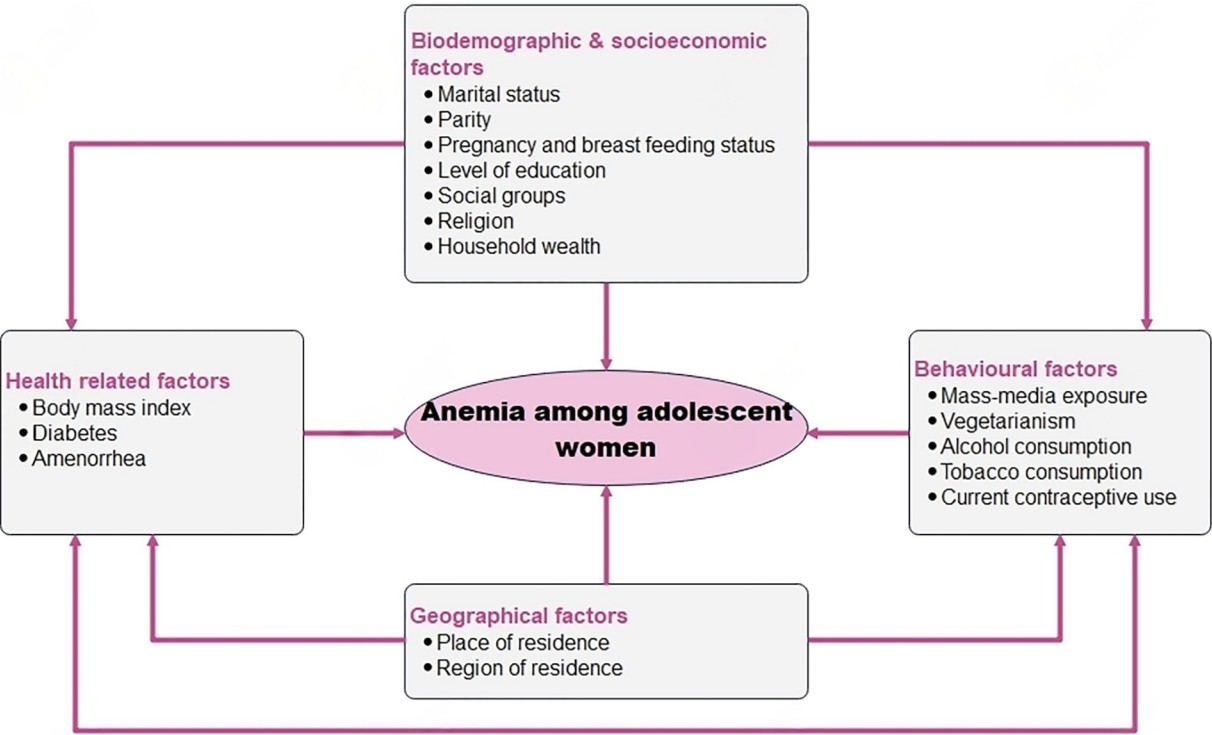

**Fig 2. Conceptual framework for factors associated with anaemia among adolescent women in India.** Source: Authors' own creation.

could gain insights into any variations or trends in anaemia prevalence at a more granular level.

To examine the independent effects of factors associated with anaemia, a multivariable binary logistic regression analysis was performed. This choice of analysis was appropriate as the outcome variable used in the analysis was dichotomous in nature. To increase the statistical power and sample size, we pooled the data from two rounds of the National Family Health Survey (NFHS), namely NFHS-4 and NFHS-5. By combining these datasets, we were also able to examine the independent effect of the survey year on the likelihood of anaemia.

Before running the multivariable binary logistic regression model, unadjusted odds ratios (ORs) along with 99% confidence intervals (CIs) were calculated to determine the significance and strength of one-to-one association between each independent variable and the outcome. This process allowed us to identify variables that exhibited a statistically significant bivariate relationship with the outcome. Variables that demonstrated a statistically significant association with the dependent variable were then included in the subsequent multivariable logistic regression analysis. By employing this approach, we ensured that only relevant predictors showing a one-to-one statistically significant relationship with the outcome were considered in the final regression model.

Since our study included several explanatory variables that might be correlated with each other, we calculated Variance Influence Factors (VIF) before putting them into pooled logistic regression analysis, to assess the multicollinearity among the independent variables. Multicollinearity among the regressors was not a problem in this study as all the VIFs were well below the recommended cut-off 5 (see S1 Table) [48].

In our analysis, we assessed the fit of the final binary logistic regression model by considering several indicators. Firstly, we examined the pseudo-R-squared value, which provides an estimate of the proportion of variance explained by the predictors in the model [49]. Furthermore, we conducted the Hosmer-Lemeshow goodness-of-fit test [50]. This test assesses the agreement between the observed and predicted outcomes and provides a measure of how well our model fits the data. A non-significant result suggests a good fit between the model and the observed data. Additionally, we evaluated the log likelihood, which measures the likelihood of obtaining the observed data based on the fitted model. A lower log likelihood indicates a better fit between the model and the data [51].

Since the surveys included in the analysis used a complex sampling design, we used the Stata command '*svyset*' to account for this design in our analysis. All the analysis was conducted in Stata16 and the maps were prepared in QGIS [52].

## Results

### Sample characteristics

Around 10% and 8% of sampled women were married before 18 years in NFHS-4 and NFHS-5, respectively. More than 80% of the sampled women in NFHS-5 were unmarried and over 90% of adolescent women did not have any children in both rounds of NFHS (Table 2). Over 90% of women in both rounds were neither pregnant nor breastfeeding. Most of the women (around 80%) were educated up to the secondary level. Around 22% (NFHS-4) to 25% (NFHS-5) of all women belong to the scheduled caste social group. More than 80% of women were Hindu. Close to 70% of the sampled women lived in rural areas. About 15–20% of women had no exposure to mass media in both rounds of the survey. A little over 70% of women were non-vegetarian and around 40% of the sampled women were of a BMI of less than 18.5. A little over 1% of women in the sample were amenorrhoeic.

**Table 2. Background characteristics of sampled adolescent women selected for the study, India, NFHS 4 and NFHS 5.**

| Background characteristics | NFHS 4 | | NFHS 5 | |
|---|---|---|---|---|
| | Frequency (N = 116117) | % | Frequency (N = 109400) | % |
| *Biodemographic and socioeconomic factors* | | | | |
| **Marital status** | | | | |
| Not married | 97,400 | 83.9 | 95,140 | 87.0 |
| Married before 18 years | 11,172 | 9.6 | 8,150 | 7.5 |
| Married 18 years and above | 7,545 | 6.5 | 6,110 | 5.6 |
| **Parity** | | | | |
| No child | 1,10,285 | 95.0 | 1,04,724 | 95.7 |
| Single child | 5,073 | 4.4 | 4,138 | 3.8 |
| 2 and more children | 759 | 0.7 | 538 | 0.5 |
| **Pregnancy and breastfeeding status** | | | | |
| Pregnant | 3,719 | 3.2 | 2,935 | 2.7 |
| Breastfeeding | 4,577 | 3.9 | 3,764 | 3.4 |
| Neither pregnant nor breastfeeding | 1,07,821 | 92.9 | 1,02,701 | 93.9 |
| **Level of education** | | | | |
| No education | 7,971 | 6.9 | 4,748 | 4.3 |
| Primary | 8,180 | 7.0 | 5,640 | 5.2 |
| Secondary | 92,081 | 79.3 | 90,936 | 83.1 |
| Higher | 7,885 | 6.8 | 8,076 | 7.4 |
| **Social groups** | | | | |
| SC | 25,964 | 22.4 | 26,826 | 24.5 |
| ST | 11,547 | 9.9 | 11,221 | 10.3 |
| OBC | 54,031 | 46.5 | 50,675 | 46.3 |
| Others | 24,575 | 21.2 | 20,678 | 18.9 |
| **Religion** | | | | |
| Hindu | 93,776 | 80.8 | 90,436 | 82.7 |
| Muslim | 16,705 | 14.4 | 14,279 | 13.1 |
| Christian | 2,255 | 1.9 | 2,210 | 2.0 |
| Others | 3,381 | 2.9 | 2,475 | 2.3 |
| **Household wealth index** | | | | |
| Poorest | 24,471 | 21.1 | 24,311 | 22.2 |
| Poorer | 26,227 | 22.6 | 25,219 | 23.1 |
| Middle | 24,871 | 21.4 | 23,346 | 21.3 |
| Richer | 22,287 | 19.2 | 20,480 | 18.7 |
| Richest | 18,260 | 15.7 | 16,045 | 14.7 |
| *Geographical factors* | | | | |
| **Place of residence** | | | | |
| Urban | 34,362 | 29.6 | 29,707 | 27.2 |
| Rural | 81,755 | 70.4 | 79,693 | 72.9 |
| **Region of residence** | | | | |
| North | 15,102 | 13.0 | 15,589 | 14.3 |
| Central | 35,037 | 30.2 | 32,898 | 30.1 |
| East | 27,181 | 23.4 | 26,906 | 24.6 |
| West | 15,237 | 13.1 | 13,144 | 12.0 |
| Southern | 20,276 | 17.5 | 18,127 | 16.6 |
| North-east | 3,285 | 2.8 | 2,737 | 2.5 |
| *Behavioural factors* | | | | |

*(Continued)*

**Table 2.** (Continued)

| Background characteristics | NFHS 4 | | NFHS 5 | |
|---|---|---|---|---|
| | Frequency (N = 116117) | % | Frequency (N = 109400) | % |
| **Mass media exposure** | | | | |
| No | 18,095 | 15.6 | 21,072 | 19.3 |
| Low | 44,011 | 37.9 | 70,065 | 64.1 |
| Medium | 46,153 | 39.8 | 18,262 | 16.7 |
| High | 7,858 | 6.8 | 0 | 0.0 |
| **Vegetarianism** | | | | |
| Vegetarian | 32,043 | 27.6 | 27,479 | 25.1 |
| Non-vegetarian | 84,074 | 72.4 | 81,921 | 74.9 |
| **Alcohol consumption** | | | | |
| No | 1,15,552 | 99.5 | 1,09,176 | 99.8 |
| Yes | 565 | 0.5 | 224 | 0.2 |
| **Consumption of tobacco in any form** | | | | |
| No tobacco | 1,14,387 | 98.5 | 1,08,489 | 99.2 |
| Uses tobacco: smoke or smokeless | 1,730 | 1.5 | 911 | 0.8 |
| **Current contraceptive use** | | | | |
| Not using | 1,13,434 | 97.7 | 1,05,022 | 96.0 |
| Traditional method | 854 | 0.7 | 1,379 | 1.3 |
| Modern method | 1,828 | 1.6 | 2,996 | 2.7 |
| *Health-related factors* | | | | |
| **Body mass index** | | | | |
| Underweight | 48,115 | 41.4 | 43,171 | 39.5 |
| Normal weight | 63,044 | 54.3 | 60,238 | 55.1 |
| Overweight | 4,021 | 3.5 | 4,623 | 4.2 |
| Obese | 937 | 0.8 | 1,368 | 1.3 |
| **Has diabetes** | | | | |
| No | 1,14,956 | 99.0 | 1,08,316 | 99.0 |
| Yes | 361 | 0.3 | 377 | 0.3 |
| Don't know | 800 | 0.7 | 706 | 0.7 |
| **Currently amenorrhoeic** | | | | |
| No | 1,14,322 | 98.5 | 1,08,128 | 98.8 |
| Yes | 1,795 | 1.6 | 1,272 | 1.2 |

Note: All percentages have been calculated using women's national weight provided in NFHS (variable V005)

## Prevalence of anaemia among adolescent women by background characteristics

The prevalence of anaemia was higher among those adolescent women who were married before 18 years. Those women who had two or more than two children had higher prevalence of anaemia than those without a child (Table 3). The prevalence was higher among those who had children and were breastfeeding than those who did not have any children and were not breastfeeding. Women who had no formal education had slightly higher prevalence of anaemia than those who were educated. The prevalence of anaemia was slightly higher among SC and ST women as compared with the rest of the social groups. About two-thirds of all ST women were anaemic in NFHS-5, slightly higher than NFHS-4. The prevalence of anaemia had slightly increased across religions between NFHS-4 and NFHS-5.

**Table 3. Prevalence of anaemia among adolescent women in India, NFHS 4 and NFHS 5.**

| Background characteristics | NFHS 4 | NFHS 5 |
|---|---|---|
| | Prevalence of anaemia (%) [99% CI] | Prevalence of anaemia (%) [99% CI] |
| | (N = 116117) | (N = 109400) |
| **India** | **54.2 [53.6,54.8]** | **58.9 [58.3,59.5]** |
| *Biodemographic and socioeconomic factors* | | |
| **Marital status** | **χ2 = 59.45, p-value: <0.001** | **χ2 = 142.39, p-value: <0.001** |
| Not married | 53.7 [53.1,54.4] | 58.3 [57.7,59.0] |
| Married before 18 years | 57.2 [55.4,58.9] | 65.1 [63.1,67.0] |
| Married on 18 years and above | 56.1 [53.9,58.2] | 59.8 [57.7,62.0] |
| **Parity** | **χ2 = 117.32 p-value: <0.001** | **χ2 = 140.15, p-value: <0.001** |
| No child | 53.8 [53.2,54.5] | 58.6 [57.9,59.2] |
| Single child | 61.2 [58.6,63.8] | 67.6 [64.9,70.2] |
| 2 and more children | 60.2 [53.5,66.6] | 63.8 [56.1,70.9] |
| **Breastfeeding and pregnancy status** | **χ2 = 111.34, p-value: <0.001** | **χ2 = 104.91, p-value: <0.001** |
| Pregnant | 52.5 [49.4,55.6] | 56.6 [53.4,59.8] |
| Breastfeeding | 61.7 [59.0,64.4] | 66.8 [63.9,69.6] |
| Neither pregnant nor breastfeeding | 54.0 [53.3,54.6] | 58.7 [58.1,59.3] |
| **Level of education** | **χ2 = 202.12, p-value: <0.001** | **χ2 = 211.67, p-value: <0.001** |
| No education | 59.8 [58.0,61.6] | 64.5 [62.2,66.7] |
| Primary | 58.3 [56.4,60.2] | 62.9 [60.8,65.0] |
| Secondary | 53.6 [53.0,54.3] | 58.9 [58.3,59.6] |
| Higher | 51.0 [48.6,53.3] | 53.1 [51.0,55.2] |
| **Social groups** | **χ2 = 307.70, p-value: <0.001** | **χ2 = 488.74, p-value: <0.001** |
| SC | 56.4 [55.2,57.7] | 61.1 [59.9,62.2] |
| ST | 60.2 [58.4,61.9] | 67.1 [65.5,68.6] |
| OBC | 53.0 [52.1,53.8] | 56.6 [55.8,57.5] |
| Others | 51.8 [50.3,53.2] | 57.3 [55.9,58.8] |
| **Religion** | **χ2 = 71.06, p-value: <0.001** | **χ2 = 61.83, p-value: <0.001** |
| Hindu | 54.7 [54.0,55.4] | 59.3 [58.6,60.0] |
| Muslim | 52.3[50.6,53.9] | 56.9 [55.2,58.6] |
| Christian | 47.9 [43.7,52.1] | 53.8 [50.0,57.5] |
| Others | 54.6 [51.1,57.9] | 61.7 [58.8,64.6] |
| **Household wealth index** | **χ2 = 353.10, p-value: <0.001** | **χ2 = 599.78, p-value: <0.001** |
| Poorest | 58.4 [57.3,59.6] | 64.0 [62.9,65.1] |
| Poorer | 55.1 [54.0,56.2] | 60.5 [59.4,61.6] |
| Middle | 54.0 [52.6,55.2] | 58.6 [57.5,59.8] |
| Richer | 52.2 [50.8,53.5] | 56.0 [54.6,57.3] |
| Richest | 50.0 [48.2,51.8] | 52.9 [51.2,54.6] |
| *Geographical factors* | | |
| **Place of residence** | **χ2 = 45.14, p-value: <0.001** | **χ2 = 108.76, p-value: <0.001** |
| Urban | 52.7 [51.3,54.1] | 56.4 [55.0,57.8] |
| Rural | 54.8 [54.2,55.5] | 59.9 [59.2,60.5] |
| **Region of residence** | **χ2 = 707.92, p-value: <0.001** | **χ2 = 1396.62, p-value: <0.001** |
| North | 52.7 [51.3,54.0] | 58.7 [57.4,60.0] |
| Central | 52.9 [52.2,53.6] | 54.6 [53.6,55.6] |
| East | 60.6 [59.4,61.9] | 67.3 [66.1,68.5] |
| West | 52.0 [49.7,54.2] | 61.4 [59.2,63.6] |
| Southern | 52.4 [50.7,54.2] | 52.3 [50.7,53.9] |

*(Continued)*

**Table 3.** (Continued)

| Background characteristics | NFHS 4 | NFHS 5 |
|---|---|---|
| | Prevalence of anaemia (%) [99% CI] | Prevalence of anaemia (%) [99% CI] |
| | (N = 116117) | (N = 109400) |
| **India** | **54.2 [53.6,54.8]** | **58.9 [58.3,59.5]** |
| North-east | 43.2 [41.1,45.3] | 61.0 [58.9,63.1] |
| *Behavioural factors* | | |
| **Mass media exposure** | **χ2 = 208.19, p-value: <0.001** | **χ2 = 177.13, p-value: <0.001** |
| No | 57.5 [56.2,58.8] | 61.7 [60.5,62.9] |
| Low | 55.3 [54.4,56.2] | 59.1 [58.4,59.8] |
| Medium | 52.6 [51.6,53.6] | 55.1 [53.6,56.5] |
| High | 49.9 [47.4,52.4] | 0 [0.0,0.0] |
| **Vegetarianism** | **χ2 = 25.67, p-value: <0.001** | **χ2 = 7.62, p-value: 0.043** |
| Vegetarian | 53.0 [52.0,54.0] | 58.2 [57.2,59.2] |
| Non-vegetarian | 54.7 [53.9,55.4] | 59.2 [58.6,59.7] |
| **Alcohol consumption** | **χ2 = 19.51, p-value: 0.019** | **χ2 = 0.39, p-value: 0.606** |
| No | 54.3 [53.6,54.9] | 58.9 [58.3,59.5] |
| Yes | 45.0 [35.1,55.2] | 61.0 [50.4,70.6] |
| **Consumption of tobacco in any form** | **χ2 = 1.91, p-value: 0.306** | **χ2 = 39.10, p-value: <0.001** |
| No Tobacco | 54.2 [53.6,54.8] | 58.8 [57.2,59.2] |
| Uses tobacco: smoke or smokeless | 55.9 [51.7,59.9] | 69.1 [64.4,73.4] |
| **Current contraceptive use** | **χ2 = 40.82, p-value: <0.001** | **χ2 = 114.29, p-value: <0.001** |
| Not using | 54.1 [53.5,54.7] | 58.6 [58.1,59.2] |
| Traditional method | 64.7 [59.2,69.8] | 69.0 [64.4,73.2] |
| Modern method | 56.0 [51.0,60.9] | 65.4 [62.1,68.6] |
| *Health-related factors* | | |
| **Body mass index** | **χ2 = 223.81, p-value: <0.001** | **χ2 = 215.02, p-value: <0.001** |
| Underweight | 56.0 [55.1,56.8] | 60.7 [59.8,61.6] |
| Normal weight | 53.6 [52.7,54.4] | 58.4 [57.6,59.2] |
| Overweight | 46.3 [43.0,49.6] | 51.2 [48.2,54.2] |
| Obese | 42.4 [35.5,49.8] | 50.8 [45.5,56.1] |
| **Has diabetes** | **χ2 = 1.08, p-value: 0.768** | **χ2 = 44.07, p-value: <0.001** |
| No | 54.2 [53.6,54.8] | 58.8 [58.2,59.4] |
| Yes | 55.1 [44.5,65.3] | 60.5 [52.2,68.3] |
| **Currently amenorrhoeic** | **χ2 = 21.18, p-value: <0.001** | **χ2 = 35.24, p-value: <0.001** |
| No | 54.1 [53.5,54.8] | 58.8 [58.2,59.4] |
| Yes | 59.6 [55.1,63.9] | 67.1 [62.1,71.7] |

Note: N = sample size, CI = confidence intervals, all percentages have been calculated using women's national weight provided in NFHS (variable V005)

The prevalence of anaemia varied across the household wealth quintiles. It was more concentrated in the lowest quintile. Adolescent women in rural areas were slightly more prone to anaemia than their urban counterparts. Anaemia was more prevalent among those women who had no mass media exposure than those who had. Women who used traditional methods of contraceptives were more anaemia than those who did other methods or did use any. The prevalence of anaemia decreased with an increase in BMI. Adolescent women who had diabetes or were experiencing amenorrhea had higher anaemia prevalence than those who had no diabetes or amenorrhea.

### Variation in anaemia prevalence across regions and states of India

The prevalence of anaemia among adolescent women in India was 54% in 2015–16 and increased to 59% in 2019–21. However, there were significant variations in the prevalence across different regions during both time periods. In 2015–16, the northeastern region had the lowest prevalence of anaemia, while the eastern region had the highest. The prevalence in the remaining regions showed minimal variation. In contrast, the prevalence of anaemia varied significantly across regions in 2019–21. It is important to note that regional estimates may mask the disparities that exist within states. Therefore, we also examined how the prevalence of anaemia varied across states in both NFHS-4 and NFHS-5.

During NFHS-4 (2015–16), certain states had notably high rates of anaemia among adolescent women. Jharkhand, West Bengal, Haryana, Bihar, and Andhra Pradesh reported the highest prevalence rates, ranging from 61% to 65%. Conversely, northeastern states such as Manipur, Mizoram, and Nagaland had the lowest prevalence, with less than 30% of adolescent women being anaemic (see Fig 3A).

In NFHS-5, the prevalence of anaemia continued to vary across states. West Bengal, Gujarat, Assam, Tripura, Jharkhand, Bihar, and Odisha reported the highest prevalence rates, ranging from 66% to 72%. On the other hand, Manipur, Kerala, Nagaland, and Mizoram reported the lowest prevalence rates, ranging from 27% to 35% (See Fig 3B).

### Change in the prevalence of anaemia during 2015–21

The overall prevalence of anaemia among adolescent women witnessed an increase from 54% during NFHS-4 to 59% in NFHS-5, with 21 states experiencing a rise in prevalence (see Fig 4).

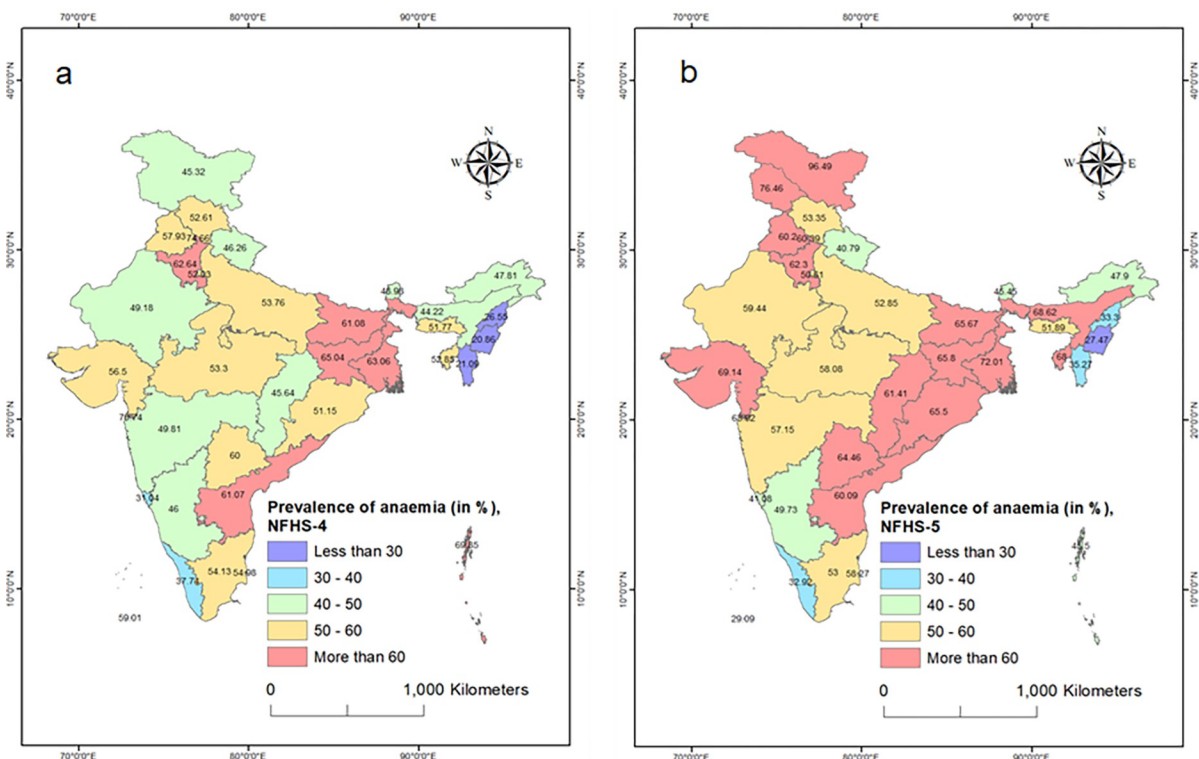

**Fig 3.** **a** and **b**: State-wise prevalence of anaemia among adolescent women in India between 2015–16 and 2019–21. All the maps used are the authors' own creations. As far as the base layer is concerned, we used a shapefile file freely downloadable from https://spatialdata.dhsprogram.com/boundaries/#view=table&countryId=IA for national and sub-national boundaries.

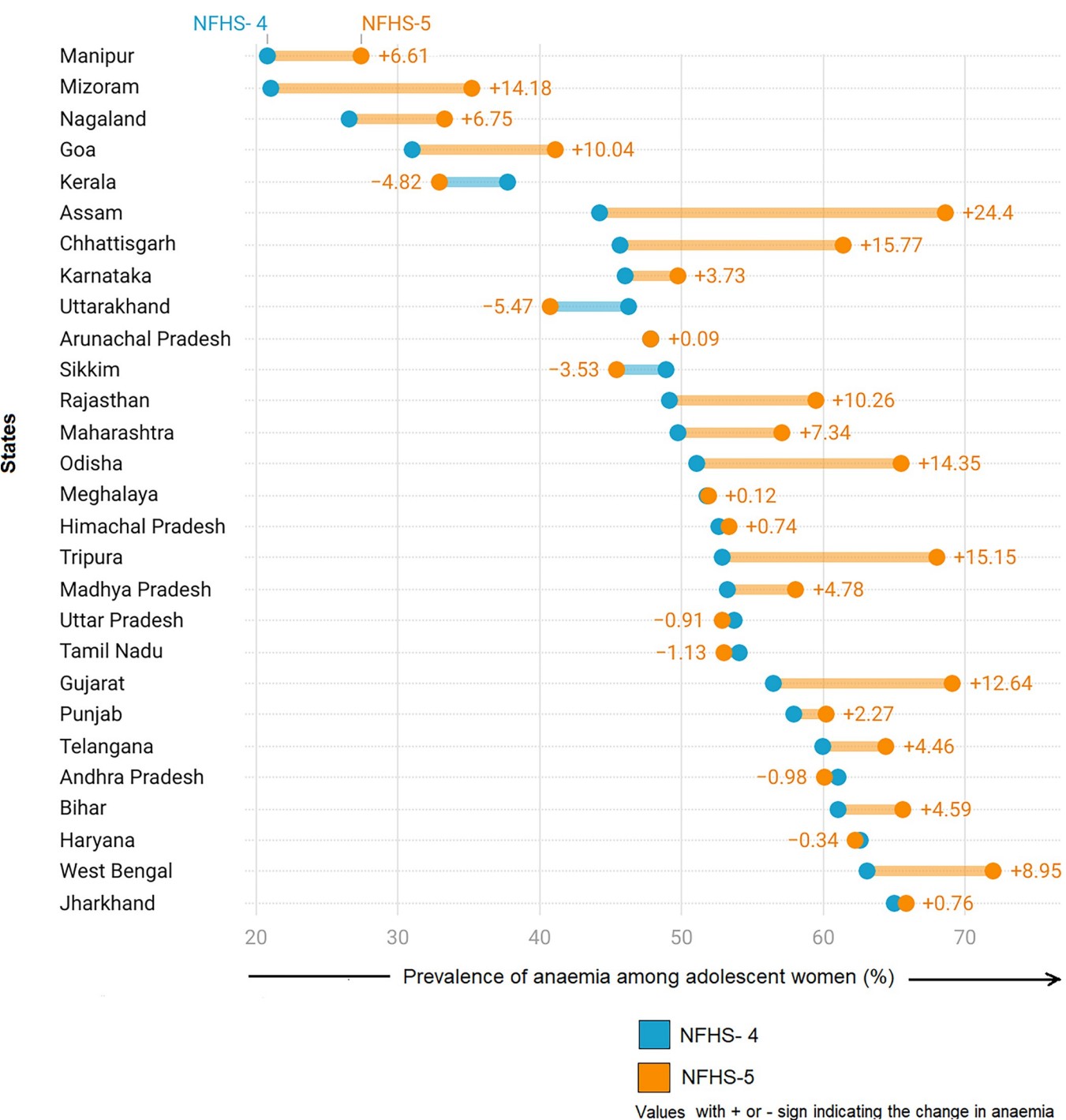

**Fig 4. Change in the prevalence of anaemia among adolescent women across the states of India between 2015–16 and 2019–21.** Source: Authors' own creation.

The magnitude of this increase varied considerably across the states. Notably, Assam, Chhattisgarh, and Tripura saw an increase of 24, 16, and 15 percentage points (pp) respectively. Similarly, Odisha, Mizoram, Gujarat, Rajasthan, and Goa witnessed a rise of 10–15 pp. West

Bengal, Maharashtra, Nagaland, and Manipur experienced a change of 5–10 pp. Nine states had only a marginal increase of less than 5 pp.

On the other hand, Uttarakhand, Kerala, Sikkim, Tamil Nadu, Andhra Pradesh, Uttar Pradesh, and Haryana recorded a decline in the prevalence of anaemia over the study period. However, this decline was marginal, amounting to about 5 pp or less.

It is worth noting that there was a significant increase in the number of states with anaemia prevalence above 60% during the study period. In 2015–16, there were already five states with prevalence rates exceeding 60%: Bihar, Jharkhand, West Bengal, Haryana, and Andhra Pradesh. However, by 2019–21, this number nearly doubled to a total of 11 states. The newly added states to this category were Gujarat, Odisha, Chhattisgarh, Assam, Punjab, Tripura, and Telangana. Remarkably, only four states consistently maintained a prevalence below 40% in both NFHS rounds, namely Manipur, Kerala, Nagaland, and Mizoram.

## Results of the logistic regression

Adolescent women who had children had slightly higher odds of being anaemic (AOR: 1.33, 99% CI: 1.16–1.51) than those without children. Adolescent women with no formal education had 25% higher odds (AOR: 1.25, 99% CI: 1.16–1.34) of having anaemia than higher educated women. The odds of anaemia were increased by 47% among ST women (AOR: 1.47, 99% CI: 1.40–1.53) than other women. The odds of being anaemic decreased with increasing household wealth. Adolescent women from the poorest wealth quintile were 17% more likely (AOR: 1.17, 99% CI: 1.12–1.23) to be anaemic than those from the richest wealth quintile. The odds of being anaemic was 24% higher among those adolescent women who lived in the eastern region (AOR: 1.24, 99% CI: 1.19–1.29) than those who lived in the northern region. Other the other hand, the odds of being anaemic were 36% lower among adolescent women living in the north-eastern region (AOR: 0.64, 99% CI: 0.61–0.67) than those living in the northern region of the country. Adolescent women who were underweight were 10% more likely (AOR: 1.10, 99% CI: 1.07–1.12) to be anaemic as compared to normal weight women. Women who were not overweight or obese were associated with reduced odds. The odds of being anaemic among adolescent women in NFHS-5 were 26% higher than those in NFHS-4 (see Table 4).

## Discussion

In recent years, anaemia among adolescent women in India has emerged as a significant public health concern, as evidenced by the NFHS-4 and NFHS-5 reports [40, 41]. Hence, this study focused on assessing the changes in anaemia prevalence across different states from 2015–16 to 2019–21. The findings of the study revealed a marginal increase in the prevalence of anaemia among adolescent women in India, from 54% in 2015–16 to 59% in 2019–21. Among the 28 Indian states, 21 states demonstrated an upward trend in anaemia prevalence, albeit to varying degrees. States such as Assam, Chhattisgarh, and Tripura exhibited a substantial rise of anaemia prevalence, more than 15 pp, while states including Punjab, Karnataka, Telangana, Bihar, and Madhya Pradesh recorded a marginal increase of less than 5 pp. Notably, a few states such as Uttarakhand and Kerala displayed a decline in anaemia prevalence, which offers potential insights for further exploration. However, a concerning observation from the study was the doubling of the number of states with anaemia prevalence exceeding 60%, from 5 in 2015–16 to 11 in 2019–21. This highlights the urgent need for targeted interventions to address this escalating issue.

To comprehend the underlying factors associated with anaemia among adolescent women, logistic regression analysis was employed. The analysis yielded noteworthy associations. Factors such as having children, belonging to a lower wealth quintile, residing in the eastern

**Table 4. Unadjusted and adjusted odds ratios (with 99% CI) for anaemia among adolescent women in India, 2015–2021 (N = 225517).**

| Background characteristics | UOR [99% CI] | p-value | AOR [99% CI] | p-value |
|---|---|---|---|---|
| *Biodemographic and socioeconomic factors* | | | | |
| **Marital status** | | | | |
| Not married (ref) | 1.00 | | 1.00 | |
| Married before 18 years | 1.21 [1.16,1.26] | <0.001 | 1.01 [0.95,1.07] | 0.747 |
| Married on 18 years and above | 1.12 [1.07,1.18] | <0.001 | 1.07 [1.01,1.14] | 0.001 |
| **Parity** | | | | |
| No child (ref) | 1.00 | | 1.00 | |
| Single child | 1.39 [1.31,1.48] | <0.001 | 1.33 [1.16,1.51] | <0.001 |
| 2 and more children | 1.32 [1.13,1.53] | <0.001 | 1.24 [1.01,1.51] | 0.006 |
| **Breastfeeding and pregnancy status** | | | | |
| Neither pregnant nor breastfeeding (ref) | 1.00 | | 1.00 | |
| Pregnant | 0.92 [0.86,0.98] | 0.001 | 0.80 [0.74,0.87] | <0.001 |
| Breastfeeding | 1.40 [1.31,1.49] | <0.001 | 0.95 [0.82,1.09] | 0.316 |
| **Level of education** | | | | |
| Higher (ref) | 1.00 | | 1.00 | |
| No education | 1.53 [1.44,1.64] | <0.001 | 1.25 [1.16,1.34] | <0.001 |
| Primary | 1.37 [1.29,1.46] | <0.001 | 1.22 [1.14,1.30] | <0.001 |
| Secondary | 1.15 [1.10,1.21] | <0.001 | 1.08 [1.03,1.13] | <0.001 |
| **Social groups** | | | | |
| Others (ref) | 1.00 | | 1.00 | |
| SC | 1.27 [1.23,1.31] | <0.001 | 1.18 [1.14,1.23] | <0.001 |
| ST | 1.18 [1.14,1.22] | <0.001 | 1.47 [1.40,1.53] | <0.001 |
| OBC | 1.10 [1.07,1.13] | <0.001 | 1.05 [1.02,1.08] | <0.001 |
| **Religion** | | | | |
| Hindu (ref) | 1.00 | | 1.00 | |
| Muslim | 0.94 [0.91,0.97] | <0.001 | 1.01 [0.98,1.05] | 0.397 |
| Christian | 0.53 [0.51,0.56] | <0.001 | 0.61 [0.57,0.64] | <0.001 |
| Others | 0.95 [0.90,1.01] | 0.020 | 0.98 [0.92,1.04] | 0.313 |
| **Household wealth index** | | | | |
| Richest (ref) | 1.00 | | 1.00 | |
| Poorest | 1.53 [1.47,1.59] | <0.001 | 1.17 [1.12,1.23] | <0.001 |
| Poorer | 1.24 [1.19,1.28] | <0.001 | 1.08 [1.04,1.13] | <0.001 |
| Middle | 1.17 [1.13,1.21] | <0.001 | 1.08 [1.04,1.13] | <0.001 |
| Richer | 1.11 [1.07,1.15] | <0.001 | 1.07 [1.02,1.11] | <0.001 |
| *Geographical factors* | | | | |
| **Place of residence** | | | | |
| Urban (ref) | 1.00 | | 1.00 | |
| Rural | 1.20 [1.17,1.23] | <0.001 | 1.03 [1.00,1.06] | 0.013 |
| **Region of residence** | | | | |
| North (ref) | 1.00 | | 1.00 | |
| Central | 0.92 [0.89,0.95] | <0.001 | 0.87 [0.84,0.90] | <0.001 |
| East | 1.35 [1.30,1.40] | <0.001 | 1.24 [1.19,1.29] | <0.001 |
| West | 1.07 [1.02,1.12] | <0.001 | 1.02 [0.97,1.07] | 0.291 |
| Southern | 0.83 [0.80,0.87] | <0.001 | 0.87 [0.83,0.90] | <0.001 |
| North-east | 0.61 [0.58,0.63] | <0.001 | 0.64 [0.61,0.67] | <0.001 |
| *Behavioural factors* | | | | |
| **Mass Media exposure** | | | | |

*(Continued)*

**Table 4.** (Continued)

| Background characteristics | UOR [99% CI] | p-value | AOR [99% CI] | p-value |
|---|---|---|---|---|
| High (ref) | 1.00 | | 1.00 | |
| No | 1.58 [1.48,1.70] | <0.001 | 1.08 [1.00,1.17] | 0.006 |
| Low | 1.40 [1.31,1.49] | <0.001 | 1.08 [1.01,1.16] | 0.004 |
| Medium | 1.16 [1.08,1.24] | <0.001 | 1.05 [0.98,1.13] | 0.055 |
| **Vegetarianism** | | | | |
| Vegetarian (ref) | 1.00 | | | |
| Non-vegetarian | 0.99 [0.97,1.01] | 0.254 | | |
| **Alcohol consumption** | | | | |
| No (ref) | 1.00 | | 1.00 | |
| Yes | 0.79 [0.70,0.90] | <0.001 | 0.95 [0.83,1.08] | 0.289 |
| **Consumption of tobacco in any form** | | | | |
| No tobacco (ref) | 1.00 | | 1.00 | |
| Uses tobacco: smoke or smokeless | 0.82 [0.76,0.88] | <0.001 | 0.99 [0.92,1.07] | 0.734 |
| **Current contraceptive use** | | | | |
| Not using (ref) | 1.00 | | 1.00 | |
| Traditional method | 1.50 [1.33,1.70] | <0.001 | 1.18 [1.03,1.34] | 0.001 |
| Modern method | 1.23 [1.13,1.33] | <0.001 | 0.99 [0.09,1.08] | 0.778 |
| *Health related factors* | | | | |
| **Body mass index** | | | | |
| Normal weight (ref) | 1.00 | | 1.00 | |
| Underweight | 1.15 [1.13,1.18] | <0.001 | 1.10 [1.07,1.12] | <0.001 |
| Overweight | 0.75 [0.70,0.79] | <0.001 | 0.78 [0.74,0.83] | <0.001 |
| Obese | 0.70 [0.62,0.79] | <0.001 | 0.73 [0.64,0.82] | <0.001 |
| **Has diabetes** | | | | |
| No (ref) | 1.00 | | | |
| Yes | 1.10 [0.90,1.34] | 0.219 | | |
| **Currently amenorrhoeic** | | | | |
| No (ref) | 1.00 | | 1.00 | |
| Yes | 1.41 [1.28,1.56] | <0.001 | 1.03 [0.91,1.17] | 0.517 |
| **Year of survey** | | | | |
| 2015–16 (ref) | 1.00 | | 1.00 | |
| 2019–21 | 1.27 [1.24,1.30] | <0.001 | 1.26 [1.23,1.29] | <0.001 |
| **Tests for model fit** | | | | |
| Log likelihood | | | -152046.32 | <0.001 |
| Pseudo R$^2$ | | | 0.02 | |
| Hosmer-Lemeshow chi$^2$ | | | 0 | 1.000 |

Note: ref = reference category, N = sample size, UOR = unadjusted odds ratio, AOR = adjusted odds ratio

region, and being ST were found to increase the odds of anaemia. Conversely, higher education, residing in the north-eastern region, and not being underweight were associated with reduced odds of anaemia.

The level of education turned out to be a significant predictor of anaemia among adolescent women. Adolescent women with higher education are less likely to be anaemic than those without formal education. This finding is supported by previous research that has shown that education is positively associated with knowledge of nutrition and health, which in turn can lead to better dietary practices and improved anaemia prevention [31, 32, 43]. Education level

also affects employment opportunities and income, leading to better access to healthcare and nutritious food, which can protect against anaemia [53].

The findings of this study indicate that adolescent women from the ST community are more likely to be anaemic than other social groups. This finding aligns with previous research indicating similar patterns [54–56]. This disparity can be attributed to various factors such as historical, socio-economic disadvantages, undernutrition, limited access to healthcare services, early childbearing, and discrimination [54]. STs have been historically disadvantaged, facing challenges in terms of education, employment, and resources [57]. These factors may contribute to their increased vulnerability to anaemia. Additionally, undernutrition and inadequate access to healthcare further exacerbate the problem [54]. Early childbearing and socio-cultural barriers can also play a role [54, 56].

The association of household wealth with anaemia is multi-faceted and can be observed through various pathways. Wealthier households tend to have better access to nutritious food, healthcare, and improved living conditions, all of which can contribute to preventing anaemia [58]. Nutrient-rich foods, which are essential for preventing anaemia, may be more accessible to wealthier households, thereby reducing the risk of developing anaemia [59]. Additionally, wealthier households may have superior sanitation infrastructure, which can help reduce infectious disease rates associated with anaemia [60, 61]. Furthermore, wealth can also affect access to healthcare, including preventative measures and treatment for anaemia. Wealthier households are more likely to have the resources to seek out and pay for healthcare, leading to better diagnosis and management of anaemia [62]. This includes access to regular check-ups, screening tests, and medical interventions to prevent and manage anaemia [63].

The relationship between BMI and anaemia among women is complex and not fully understood. However, the previous research suggests that underweight women have a higher risk of anaemia than those who are overweight or obese [64]. This may be due to a lack of adequate nutrient intake and poor overall health in underweight individuals. Additionally, research has suggested that being overweight and obese may be protective against anaemia due to increased iron stores in the body. A study by Nguyen et al, published in the Journal of Nutrition in 2012, found that women with a BMI of less than 18.5 were more likely to be anaemic than those with a BMI greater than 25 [65]. Additionally, another study found that underweight women had a significantly higher risk of anaemia compared to normal-weight women [34]. However, it is important to note that BMI alone may not be a perfect indicator of nutritional status and overall health, and there may be other factors contributing to the relationship between BMI and anaemia. Further research is needed to fully understand the pathways through which BMI is associated with anaemia among women.

Our study revealed that women from the northeastern states of India were at lower risk of anemia than women from the other parts of the country. This is likely due to the diverse and nutritious diet that is traditionally consumed in these states. The culture of the north-eastern states of India may also play a role, with many people following a traditional diet that emphasizes locally-grown and seasonal foods [66, 67]. Research revealed Bao or red rice, grown by local farmers of this region has unique nutritional value and is rich in iron, and other micronutrients [68]. This rice is very popular among northeastern people. According to National Sample Survey 2011–2012 (68th round) the red meat consumption (more than 20 grams/day) among the northeastern states was highest in the country. These are among some of the factors that may have contributed to the relatively low rates of anaemia in some of the northeastern states of India.

The Indian government has implemented several programs to address anaemia among women and children in the country over past five decades. To address the issue of maternal and infant anaemia, Ministry of Health and Family Welfare (MoHFW) established the

Maternal and Child Health (MCH) division in 1970, and initiated National Nutritional Anaemia Prophylaxis Programme (NNAPP) nation-wide. Women of childbearing age, breastfeeding mothers, and children under the age of five were given Iron and Folic Acid (IFA) supplements as part of this initiative [69]. The Integrated Child Development Services (ICDS) launched in 1975 to provide health, nutrition, and early childhood education services [70]. The MoHFW updated anaemia prevention and control recommendations in 1991 and the National Nutritional Anaemia Control Programme (NNACP) replaced the NNAPP. The national iron supplementation policy was again updated in 2007, coinciding with the launch of Weekly Iron and Folic Acid supplementation (WIFS) Program. Under this initiative, IFA supplements were given to all 6th to 12th class adolescent boys and girls enrolled in all government/government-aided schools. Subsequently, in 2013, the National Health Mission (NHM) was launched to improve health and well-being in India and included components focused on maternal and child health, including anaemia [71].

The Mid-Day Meal Scheme, launched in 1995, provides free meals to children in government-run schools. The *Poshan Abhiyaan* (English translation: Nutrition Campaign) launched in 2018 aims to address malnutrition among women and children and includes a focus on anaemia [72]. The National Iron Plus Initiative (NIPI) launched in 2013 is a program aimed specifically at addressing anaemia among women and children, providing iron and folic acid supplementation, health and nutrition education, and improving access to sanitation facilities [73]. These programs are implemented through the existing health infrastructure and are being implemented in collaboration with state governments, non-government organizations, and private sector partners. However, it is important to note that while these programs have made significant contributions to addressing anaemia, there are other factors that may influence the prevalence of anaemia, including infections and lifestyle changes that are not specifically targeted by these programs. Infections, such as parasitic infections and other infectious diseases, can contribute to anaemia by causing blood loss or impairing the body's ability to absorb and utilize iron [74]. Additionally, several lifestyle-like changes in dietary patterns, may contribute to an increased prevalence of anaemia. These factors, if not adequately addressed, can undermine the effectiveness of the existing programs and contribute to the persistence of anaemia in certain populations.

Despite the existing programs, the anaemia level among women in India, particularly among adolescent women of reproductive age the anaemia is rapidly increasing. Insufficient coverage, inadequate implementation, lack of behavioural change, inadequate monitoring and evaluation, the persistence of underlying risk factors, and resistance to good dietary practices could be some of the factors contributing to the limited success of programs aimed at reducing women's anaemia levels in India [75]. In addition to these factors, the lack of coordination between different programs and stakeholders may also be limiting the impact of these initiatives. When there is a lack of coordination, there may be duplication of effort, conflicting messages, and limited impact, making it more difficult to effectively reduce anaemia among women and children in India. Addressing these obstacles and ensuring effective coordination between different programs and stakeholders will be key to achieving progress in reducing anaemia in India.

The findings of this study offer important implications for policymakers in reducing anaemia among adolescent women in India. The study highlights the association of various factors, such as maternal and child health, education, economic status, regional differences, and nutrition, in determining the prevalence of anaemia among this population. By considering these findings, policymakers can tailor their interventions to address the anaemia problem among adolescent women in India effectively. For instance, the findings that adolescent women who had children and those with higher education have lower odds of anaemia indicate the importance of investing in maternal and child health programs, as well as programs that focus on

nutrition and health education. On the other hand, the findings that the odds of anaemia are higher among ST women compared to other women and those living in the eastern region compared to the northern region highlight the need for culturally sensitive and state-specific interventions. Additionally, the finding that adolescent women who are not underweight are less likely to be anaemic highlights the importance of addressing malnutrition through improved access to nutritious food and health services. By doing so, the government can help to ensure better health outcomes for this vulnerable population and promote health equity. The findings of this study also highlight the need for continued monitoring and evaluation of anaemia programs, such as the NIPI and the *Poshan Abhiyan*, to ensure their effectiveness in reducing anaemia among adolescent women in India.

The limitations of this study on anaemia among adolescent women in India should be noted in order to provide a more nuanced interpretation of the results. Firstly, as it is a cross-sectional study, it is not possible to establish causality between the independent variables and the outcome. Secondly, some important variables related to the intake of folate, vitamin B12, and vitamin A, access to healthcare services, and cultural beliefs and practices, which could have an impact on the prevalence of anaemia, have not been included in the study due to lack of such variables in the dataset. Additionally, anaemia can have a significant association with different types of infections. However, since NFHS does not provide any information about infections among women we could not address this issue. Finally, it is important to acknowledge that haemoglobin concentrations were measured using a battery-operated portable HemoCueHb 201+ analyzer. While this device offers the convenience of on-site testing, it is essential to recognize that its results may differ from those obtained through laboratory testing methods. The HemoCueHb 201+ analyzer has its own set of limitations and potential sources of error. Combining both portable and laboratory-based testing methods, along with appropriate quality control measures, can provide a more comprehensive assessment of haemoglobin concentrations and enhance the accuracy of diagnostic and research findings. These limitations should be considered in future research to provide a more comprehensive and accurate picture of the prevalence of anaemia among adolescent women in India.

## Conclusion

In conclusion, this study provides valuable insights into the changing prevalence of anaemia over time and the associated factors among adolescent women in India. The findings revealed that anaemia affected a large number of adolescent women in the country and there was a marginal increase in anaemia prevalence among this sub-group of population during the study period. These results emphasize the necessity for targeted interventions, particularly in states experiencing a notable rise in anaemia prevalence. Furthermore, the study highlights the crucial role of socioeconomic variables, such as wealth, education, and social group membership, in determining the risk of anaemia among adolescent women in India. It reveals that poor nutritional status and adolescent motherhood are additional risk factors for anaemia in this population. These findings underscore the urgency of addressing anaemia among vulnerable groups within this demographic. Policymakers can utilize these research findings to guide the development of tailored interventions aimed at reducing anaemia prevalence. By incorporating these insights, policymakers can enhance the effectiveness of anaemia reduction programs, nutrition education initiatives, and state-specific interventions.

## Supporting information

**S1 Table. Variance inflation factors.**
(DOCX)

## Acknowledgments

Dr. Aditya Singh acknowledges the support of Institute of Eminence Seed Grant number R/ Dev/D/IoE/Equipment/Seed Grant-II/2022-23/48726 provided by Banaras Hindu University. Mahashweta Chakrabarty acknowledges the support of Junior Research Fellowship provided by University Grants Commission (UGC).

## Author Contributions

**Conceptualization:** Aditya Singh.

**Data curation:** Mahashweta Chakrabarty, Aditya Singh.

**Formal analysis:** Mahashweta Chakrabarty, Aditya Singh.

**Investigation:** Aditya Singh, Shivani Singh.

**Methodology:** Mahashweta Chakrabarty, Aditya Singh.

**Project administration:** Aditya Singh, Shivani Singh.

**Resources:** Aditya Singh.

**Software:** Mahashweta Chakrabarty.

**Supervision:** Aditya Singh, Shivani Singh.

**Validation:** Aditya Singh, Shivani Singh.

**Visualization:** Mahashweta Chakrabarty.

**Writing – original draft:** Mahashweta Chakrabarty, Aditya Singh, Sourav Chowdhury.

**Writing – review & editing:** Aditya Singh, Shivani Singh, Sourav Chowdhury.

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
