## [Decision Letter · Decision Letter 0]

22 Mar 2023

PGPH-D-23-00195

Is the burden of anaemia increasing among Indian adolescent women? Evidence from Indian demographic and health surveys (2015-21)

Dear Dr. Chakrabarty,

Thank you for submitting your manuscript to PLOS Global Public Health. After careful consideration, we feel that it has merit but does not fully meet PLOS Global Public Health’s publication criteria as it currently stands. Therefore, we invite you to submit a revised version of the manuscript that addresses the points raised during the review process.

We look forward to receiving your revised manuscript.

Kind regards,

Rajesh Sharma, Ph.D.

Academic Editor

Journal Requirements:

1. Please send a completed 'Competing Interests' statement, including any COIs declared by your co-authors. If you have no competing interests to declare, please state "The authors have declared that no competing interests exist". Otherwise please declare all competing interests beginning with the statement "I have read the journal's policy and the authors of this manuscript have the following competing interests:"

2. Some material included in your submission may be copyrighted. According to PLOS’s copyright policy, authors who use figures or other material (e.g., graphics, clipart, maps) from another author or copyright holder must demonstrate or obtain permission to publish this material under the Creative Commons Attribution 4.0 International (CC BY 4.0) License used by PLOS journals. Please closely review the details of PLOS’s copyright requirements here: PLOS Licenses and Copyright. If you need to request permissions from a copyright holder, you may use PLOS's Copyright Content Permission form.

Potential Copyright Issues:

Figure 3 & 4: please (a) provide a direct link to the base layer of the map (i.e., the country or region border shape) and ensure this is also included in the figure legend; and (b) provide a link to the terms of use / license information for the base layer image or shapefile. We cannot publish proprietary or copyrighted maps (e.g. Google Maps, Mapquest) and the terms of use for your map base layer must be compatible with our CC-BY 4.0 license. 

Additional Editor Comments (if provided):

Major Revision

Reviewers' comments:

Reviewer's Responses to Questions

**Comments to the Author**

1. Does this manuscript meet PLOS Global Public Health’s publication criteria? Is the manuscript technically sound, and do the data support the conclusions? The manuscript must describe methodologically and ethically rigorous research with conclusions that are appropriately drawn based on the data presented.

Reviewer #1: Partly

Reviewer #2: Partly

2. Has the statistical analysis been performed appropriately and rigorously?

Reviewer #1: No

Reviewer #2: Yes

3. Have the authors made all data underlying the findings in their manuscript fully available (please refer to the Data Availability Statement at the start of the manuscript PDF file)?

Reviewer #1: Yes

Reviewer #2: No

4. Is the manuscript presented in an intelligible fashion and written in standard English?

Reviewer #1: Yes

Reviewer #2: No

5. Review Comments to the Author

Reviewer #1: 3/2/2023

Review of “Is the burden of anaemia increasing among Indian adolescent women? Evidence from Indian demographic and health surveys (2015-21).”

Thank you for the opportunity to review this informative manuscript.

The manuscript describes a study that focuses on the prevalence of anemia among adolescents in India using two demographic health surveys from 2015 and 2021. The study’s goals were to measure the change in the overall prevalence in the country and to explore factors that may influence anemia rates. The main findings include the increased anemia prevalence and several risk and protective factors. The authors provide their interpretations of the overall and specific findings and suggest policy directions toward better control of anemia.

This manuscript deals with one of the most important and persistent global public health problems. Anemia has drawn so much public health efforts over decades. The finding that anemia is on the rise among adolescents in nationally representative samples has startling implications for India and many other countries. As such, this manuscript has exciting potential to become an important contribution to PLOS Global Public Health. Having said this, this manuscript is underdeveloped in several regards, including the conceptual framework, analytic approach, and the presentation and discussion of results.

Main comments:

I would like to see a careful revision on the definition of anemia. The paper focuses on anemia of low hemoglobin concentration, rather than “deficient” hemoglobin, which implies hemoglobin polymorphisms like sickle-cell trait, which can also result in anemia. You do mention the importance of iron and other micronutrients and (to a lesser extent) infection. So, it is implicit that this study is concerned with anemia of diverse types.

I feel that the manuscript could be strengthened by paying more attention to anemia arising from infection. Low hemoglobin could occur as part of immune response to infections or disease processes. Even though nutritional (particularly iron deficiency) anemia is usually the most prevalent, other types of anemia are also of crucial importance. This is particularly the case when one tries to understand why preventive programs focused on iron and folate supplementation have had low efficacy. Anemia of infection (either as a genetic adaptation to malaria or phenotypic response to combat infections) should be incorporated in the introduction and discussion, and ideally in method and analysis as well. It is my understanding that recent Demographic Health Survey data often include biomarkers of infection/inflammation. It might even be possible to include it in the statistical evaluation.

For the conceptual framework, it seems inadequate to place health-related factors entirely separated from geographical/demographic factors. It seems like a better idea to think of geographical and economic factors as more distal factors that may influence some health factors. One could then construct the framework where economic factors have direct and indirect effects on anemia prevalence, the latter mediated by health factors. I understand that it is necessary to simplify concepts for clarity, but treating geographical and demographic factors completely separately from health factors is unrealistic and prevents this study from addressing the interplay between factors that the authors attempt to highlight. Please see below my specific comments and suggestions.

Detailed comments

Abstract:

“Despite several government initiatives…. The burden of anaemia has increased over time” - I do not recall seeing comparable statements and references in the introduction section.

“Factors such as nutrition, parity, …. Have a role in the prevalence of anemia.” – The word “role” implies a causal influence on anemia. Please consider editing appropriately for an association study.

Introduction

Line 47 “a deficiency in red blood cells or haemoglobin” does not do justice to the definition of anemia more broadly.

Line 50 Please define “GBD.” Also, this sentence implies that anemia directly kills. Is it true? Or is anemia attributed to a distal cause for things like infectious diseases that kill people? Either way, clarifying this with sources would be important.

Line 59 Parasitic infections are mentioned here, but it does not make clear the multiple roles infectious diseases can play in malaria prevalence. Anemia due to parasitic infections (e.g., hookworm) and anemia due to the human body’s immune response (i.e., iron sequestration from infectious agents) are important causes of low circulating hemoglobin in environments of high infectious load. As currently written, this section gives the impression that anemia is caused by malnutrition, which happens to coincide geographically with high rates of parasitic infections. This minimizes the importance of infections as a direct cause of anemia. (Although it is also true that malnutrition increases the risk for infectious diseases).

Line 100 It is good to see the reference to the idea, “infections that can cause anemia” here, but the paper would be strengthened by increased attention to this type of anemia.

Line 118 “poverty which limits access to nutritious food and poor sanitation …” does not seem to make sense. Please edit.

The paragraph starting with line 125 seems to include statements that require citation of sources. Line 128, the notion that societal views that prioritize men over women leads to women’s inadequate nutritional consumption should be supported by sources, even though this makes intuitive sense.

Line 131. Please cite sources for the notion that anemia increases due to iron loss via breastmilk. This is contrary to my understanding that human milk does not draw significant amounts of iron from the mother compared to the amounts drawn during pregnancy. Iron-binding proteins like lactoferrin are abundant in milk, but they seem to serve more of immune role than iron transport based on more recent research. So, the information you provide might be outdated and therefore requires either revision or proper citation to allow the reader to evaluate the information quality.

Paragraph starting with 139. This paragraph would be strengthened if you added hypotheses or expectations, or specific aims. Currently, some of this seems to appear in your data and method section (e.g., independent variables), but it makes better sense to state your aims or hypotheses first and then you describe the methods you used to accomplish those aims.

Line 155. I understand that the sampling strategies for surveys are available elsewhere, but it would help the reader if you provided a brief description of sampling strategies for the two surveys. Were the strategies identical across two surveys?

Line 173 Please consider editing this sentence to reflect the cross-sectional research design. I am concerned about the word, “affecting” which implies causal order, which is not warranted.

Line 183 Please define what “Scheduled” means.

Line 184 “backword” classes – please either explain this or edit this expression. It sounds value-laden and offensive to me (from a global perspective). I realize this is something to do with the caste system, but it would be preferable to avoid discriminatory expressions if that is possible and appropriate.

Line 186 Please provide cutoffs for wealth index.

Line 187 Please provide information on media exposure. Is no exposure based on self-reporting? I wonder about the validity of this category – no exposure means no radio, no TV, or no newspaper? Have you considered combining no exposure with low exposure to obtain three groups instead of four?

Line 196 You mention that you used descriptive statistics for background characteristics and anemia prevalence by group categories. Did you evaluate associations/correlations among predictors? Please consider doing so and provide this information before presenting regression models.

Line 198 Is multivariable the same as multivariate? I am not familiar with the former expression. Also, you used binary logistic regression models, but your outcome variable is described as ordinal with a few cutoff levels of hemoglobin. Or did you code the variable dichotomously using different cutoffs for each set of models? Please clarify.

On the four models and determination to retain or drop some variables, please consider adding a brief discussion of your reasoning behind this sequence of modeling – why certain sets of variables were evaluated first and health variables last. This may be a conventional sequence in the authors’ fields, but it does not seem intuitive to interdisciplinary readers.

Line 204 You used the statistical significance cutoff of .05 – is this level appropriate for this large sample size? Have you considered a lower cutoff such as .01 or .001? Your sample size is quite large, and that can inflate the chances of finding statistical significance. You might have already consulted with a statistician, but if you have not, I would.

Around here, please add a statement on how the change over time was evaluated. From the logistic model table, I am guessing that you combined the two sets of survey data to run the models. This is not forthcoming in your description of the methodological approach to statistical analysis.

Please provide information on how you evaluated model fit and issues like multicolinearity.

Line 216 “either breastfeeding not pregnant” does not make sense. Please edit.

Line 218 Please define SC/ST.

Line 224 Did you distinguish amenorrhea types (pregnancy, postpartum, undernutrition)? This would be useful, here and/or in variable descriptions.

Line 23 “before the legal age of marriage” this information should come in your variable description instead of results. Also, “two or more two children” does not make sense.

Line 233 I see the expression “more anemic” here and elsewhere. It implies more severe anemia. Please consider using expression higher or lower prevalence instead.

Line 234 please replace “anemia is” with “anemia was”

Line 253 In describing the map, the main text seems to focus on the state-level changes (which state claimed the highest prevalence, etc.) and misses what I perceive as the most striking overall pattern of change - the expanded regions with high anemia prevalence in 2021.

Table 3 Please provide a caption for the table and state the model fit such as pseudo-R-squared, define the abbreviation you used for the reference groups, and the sample size.

Line 302 Please consider replacing “being overweight or obese reduce the odds” with “not underweight was associated with reduced odds.” It seems incorrect to specify overweight and obesity when normal weight is also associated with reduced odds.

Line 315 Wealthier households may also have a superior sanitation infrastructure, which may decrease infectious disease rates.

Line 333 Diversified diets are indeed protective against micronutrient deficiencies, but I am not following your idea of fruits, vegetables, and fish are the ones that help prevent anemia. These foods are unlikely to provide significant amounts of iron. Please explain how these foods are specifically beneficial. Alternatively, revise this section appropriately.

Line 345 I would be cautious not to recommend iron supplements because they can be harmful to individuals in the presence of infections. I appreciate that you specify, “as prescribed by their healthcare provider” here.

Line 361 12 gm% seems like a strange unit. Please check.

Line 376-377 The sentence is incomplete.

Line 386 Around here, you might add a discussion on infections and other lifestyle changes not addressed by these programs as factors behind the increased prevalence.

Line 388 You mention that the study highlights “the interplay of various factors” but to me study has not yet done this. The study has identified factors that are associated with anemia and you have discussed them individually but very little of the interplay. Please consider editing the sentences with this word. Better yet would be to incorporate the notion of interplay in your conceptual framework and statistical modeling so that you can address how factors may overlap, mediate, or moderate the effect of other factors to give a holistic picture. (This relates to my comments about the issues in your current conceptual framework).

Line 393 “women who had children … have lower odds. This contradicts your results reported in the logistic regression table – elevated odds. Please double check.

Conclusion paragraph – please consider editing the words “interplay” and “affecting” (for the reason explained above).

Reviewer #2: This is an interesting and thorough study looking at risk factors for anemia in India. The main strength of this study is the large sample size but not enough time is spent discussing the dataset or how the variables are defined or collected. More time needs to be spent in the main text describing the sample, the sampling method, and data collection methods. Likewise, some of the variables are ill defined. The introduction is very long and words could easily be saved here for them to be used in the methods section. The information on the adjusted variables could also perhaps be added into a supplementary table. All variables need to be better defined, but these key ones need to be for the reader to appreciate the manuscript fully:

- Were all independent variables taken from the NFHS-4 and NFHS-5 surveys?

- If different categories of anemia are being diagnoses (mild, severe, moderate) then the dependent variable is not dichotomous, it is categorical. It needs to be made clear if all 3 categories were dichotomized.

- How was household wealth index defined? The categories ‘poorest’, ’poor’,’ middle’, ’richer’, ’richest’ are not very informative. Was the sample split into quintiles based on wealth or were national level scales used?

- What is a traditional method of contraception?

- What BMI cut offs were used? Were they the south Asian specific cut offs?

It is also very important that a comment be made regarding the term ‘backwards class’. Although this term is used by the Indian census it could be seen as discriminatory. This manuscript below uses the term and then has a footnote describing this which could be good to use for guidance. https://www.frontiersin.org/articles/10.3389/fpsyg.2017.00487/full)

Statistical values in text are not adequately described. Using the words ‘around’, ‘large majority’ , ‘a little over’ and ‘over’ should be avoided and actual numbers used. Percent also does not need to be spelt out. Using terms like ‘every four out of five women’ (line 218) should be replaced with 80% or a ratio. Overall, the first paragraph of the results is hard to read as % values, ratios and written phrases are all used to describe proportions. It should be consistent. It is also not clear what a weighted % means in Table 1.

The English language (spelling, grammar and sentence structure) needs to be improved prior to publication.

Minor comments:

Abstract

- Around 54-59% of women affected in the abstract is not scientific enough

- As the manuscript is in relation to changes over time more statistics about these temporal changes need to be put in the abstract

- It is not clear from the abstract if nutrition, parity, breastfeeding status, place of residence and SES condition are statistically significantly associated with anemia

Introduction

- It is not clear from the introduction how anemia can lead to death – this may be helpful for readers less familiar with the condition

- The introduction is interesting but quite long and repetative. Some of the statistics could be taken out and the same overall message of the importance of anemia be portrayed. (Notably sections from lines 70-87, 106-115).

- Line87 needs a reference

Methods

- More information on the sampling procedure would be nice in the main text. Was every adult approached or was it a targeted sample?

- It may be nice to provide some more information into similarities and differences between datasets

- What type of blood sample (plasma or serum)? Were they fasted (lines 158-159)

- Who was eligible? (lines 159-160)

- Dietary habit should be changed in vegetarianism as other dietary habits exist.

Results

- The statewide prevalence trends appears to come out of nowhere. It should be mentioned in the methods that this was going to be done so that the manuscript is clearer for the reader to follow.

Discussion

- Line 376 is not a full sentence

Tables

- Adding in the years the NFHS were taken in the table may be helpful to the reader to appreciate why some associations were lost

- For the reference group in table 3 it may be clearer to put the OR as 1 in the table as typically seen.

6. PLOS authors have the option to publish the peer review history of their article (what does this mean?). If published, this will include your full peer review and any attached files.

**Do you want your identity to be public for this peer review?** For information about this choice, including consent withdrawal, please see our Privacy Policy.

Reviewer #1: **Yes: **Masako Fujita

Reviewer #2: No

---

## [Decision Letter · Decision Letter 1]

19 Jul 2023

Is the burden of anaemia among Indian adolescent women increasing? Evidence from Indian Demographic and Health Surveys (2015-21)

PGPH-D-23-00195R1

Dear Dr. Singh,

We are pleased to inform you that your manuscript 'Is the burden of anaemia among Indian adolescent women increasing? Evidence from Indian Demographic and Health Surveys (2015-21)' has been provisionally accepted for publication in PLOS Global Public Health.

Best regards,

Rajesh Sharma, Ph.D.

Academic Editor

Accept

Reviewer Comments (if any, and for reference):

Reviewer's Responses to Questions

**Comments to the Author**

1. If the authors have adequately addressed your comments raised in a previous round of review and you feel that this manuscript is now acceptable for publication, you may indicate that here to bypass the “Comments to the Author” section, enter your conflict of interest statement in the “Confidential to Editor” section, and submit your "Accept" recommendation.

Reviewer #2: All comments have been addressed

2. Does this manuscript meet PLOS Global Public Health’s publication criteria? Is the manuscript technically sound, and do the data support the conclusions? The manuscript must describe methodologically and ethically rigorous research with conclusions that are appropriately drawn based on the data presented.

Reviewer #2: Yes

3. Has the statistical analysis been performed appropriately and rigorously?

Reviewer #2: Yes

4. Have the authors made all data underlying the findings in their manuscript fully available (please refer to the Data Availability Statement at the start of the manuscript PDF file)?

Reviewer #2: Yes

5. Is the manuscript presented in an intelligible fashion and written in standard English?

Reviewer #2: Yes

6. Review Comments to the Author

Reviewer #2: Many thanks for addressing all comments with such rigor. This is an interesting study and contributes to the field positively

7. PLOS authors have the option to publish the peer review history of their article (what does this mean?). If published, this will include your full peer review and any attached files.

**Do you want your identity to be public for this peer review?** For information about this choice, including consent withdrawal, please see our Privacy Policy.

Reviewer #2: No
